# Metabolic Syndrome, Inflammation, Oxidative Stress, and Vitamin D Levels in Children and Adolescents with Obesity

**DOI:** 10.3390/ijms251910599

**Published:** 2024-10-01

**Authors:** Tjaša Hertiš Petek, Evgenija Homšak, Mateja Svetej, Nataša Marčun Varda

**Affiliations:** 1Department of Pediatrics, University Medical Centre Maribor, Ljubljanska ulica 5, 2000 Maribor, Slovenia; tjasa.hertispetek@ukc-mb.si; 2Department of Laboratory Diagnostics, University Medical Centre Maribor, Ljubljanska ulica 5, 2000 Maribor, Slovenia; evgenija.homsak@ukc-mb.si (E.H.); mateja.svetej@ukc-mb.si (M.S.); 3Faculty of Medicine, University of Maribor, Taborska ulica 8, 2000 Maribor, Slovenia

**Keywords:** children, inflammation, oxidative stress, metabolic syndrome, obesity, vitamin D, visceral fat thickness, I-TAC/CXCL11, mediation analysis

## Abstract

Metabolic syndrome (MetS) is associated with systemic inflammation, oxidative stress, and hypovitaminosis D. Our aim was to determine whether vitamin D mediates inflammation and oxidative stress, assessed through selected biomarkers, in children with obesity and/or MetS. Eighty children with normal weight, overweight, or obesity were analyzed for serum vitamin D, C-reactive protein, leukocytes, adiponectin, monocyte chemoattractant protein-1, myeloperoxidase, interferon-inducible T-cell alpha chemoattractant (I-TAC/CXCL11), superoxide dismutase-1, fasting lipid and glucose levels, ultrasound-measured abdominal fat thickness, waist circumference, body mass index and blood pressure. Children with obesity or overweight had lower vitamin D levels, increased blood pressure, visceral and subcutaneous fat thickness, and higher leukocytes, C-reactive protein, and myeloperoxidase levels. Those with MetS also had lower adiponectin levels. Vitamin D levels are negatively correlated with body mass index, waist circumference, and visceral and subcutaneous fat thickness. Correlation, mediation, and regression analyses showed no link between vitamin D and inflammatory/oxidative stress variables. The novel biomarker I-TAC did not correlate with obesity or vitamin D status. Our results indicate that vitamin D does not significantly mediate inflammation or oxidative stress in children and adolescents with obesity and/or MetS. Selected inflammation/oxidative stress biomarkers appear to be altered primarily due to obesity rather than vitamin D status.

## 1. Introduction

Metabolic syndrome (MetS) comprises a group of cardiovascular risk factors—arterial hypertension, altered glucose metabolism, dyslipidemia, abdominal obesity, and pro-inflammatory states [1,2,3,4] that frequently occur in children with obesity [4]. It is the result of complex interactions between central obesity, insulin resistance, low-grade inflammation, and other factors in individuals with a genetic predisposition [1,2,3,5]. While the pathogenesis of MetS is not completely understood, insulin resistance and subsequent inflammation are thought to be among its main mechanistic underpinnings [6].

Obesity, particularly central or abdominal obesity, is a central component of MetS [4]. Abdominal fat accumulation is associated with increased secretion of inflammatory cytokines and adipokines, contributing to insulin resistance and other metabolic disturbances [7,8]. Well-established biomarkers of inflammation and oxidative stress in obesity and MetS are interleukins (interleukin-1, -6, -10, -18), adiponectin, resistin, tumor necrosis factor-alpha, leptin, monocyte chemoattractant protein-1, angiotensinogen, plasminogen activator inhibitor-1, myeloperoxidase, and E-selectin, among others [9,10] (Figure 1).

Interferon-inducible T-cell alpha chemoattractant (I-TAC/CXCL11) is a novel biomarker of inflammation in obesity, belonging to the CXC chemokine family, which plays a role in the recruitment of T cells and other immune cells to sites of inflammation. In adults, it has been associated with endothelial leukocyte stasis and increased risk of cardiovascular morbidity in obesity, as well as with adipose tissue angiogenesis [13,14]. Data suggest that CXCL family members, specifically CXCL10 and CXCL11, may serve as potential biomarkers for the onset of adipose tissue inflammation during obesity [15]. Given that CXC chemokine profiling is closely linked to leukocyte and macrophage recruitment and has significant immunomodulatory potential, it may be useful in predicting the therapeutic potential for managing obesity and obesity-related diseases such as type 2 diabetes and non-alcoholic fatty liver disease [16]. However, there is limited research on I-TAC in adult obesity, and no data are currently available regarding its role in childhood obesity.

Vitamin D deficiency is increasingly prevalent on a global scale and is connected to chronic morbidity, including cardiovascular diseases, diabetes, obesity, and inflammation [17]. Recently, the extraskeletal effects of vitamin D on adipose tissue biology and its modulation in human obesity have become of great interest [18]. It is hypothesized that vitamin D impacts the hormonal regulation of glucose metabolism and the synthesis of adipokines by fat tissue [19]. Lower vitamin D levels also appear to modulate systemic cytokine production and increase white blood cell (WBC) count [20,21,22]. The mechanism by which vitamin D might mediate the association between obesity and inflammation might involve its hormonal form of vitamin D, calcitriol, which has immunomodulatory properties through the vitamin D receptor (VDR) [23] (Figure 2). A recent study proposed that low vitamin D levels in obesity lead to the activation, proliferation, and production of pro-inflammatory mediators by T cells. Hypovitaminosis D is thought to cause a decrease in the functional potential of regulatory and anti-inflammatory lymphocytes and the maintenance of the inflammatory response. It was concluded that hypovitaminosis D is a possible link between subclinical inflammation in obesity and the development of concomitant severe diseases, as well as a decrease in the functional properties of immune cells [24]. While specific studies directly linking vitamin D to I-TAC levels are, to our knowledge, lacking, vitamin D’s broader role in regulating inflammation and immune function suggests it could affect I-TAC production.

In children with severe obesity, low vitamin D levels were detected, along with increased markers of oxidative stress, inflammation, and endothelial activation [26]. An association between inflammation and obesity is evident, but the role of vitamin D in this process remains unclear [23]. Current evidence from preclinical and clinical studies in human adipose tissue supports the anti-inflammatory effects of vitamin D. However, the molecular effects of vitamin D on adipocyte differentiation, adipogenesis, energy metabolism, and the changes in adipokine levels after vitamin D supplementation are inconclusive [18].

The aim of our study was to explore some traditional inflammatory and oxidative stress markers, along with the novel inflammatory biomarker I-TAC, in relation to vitamin D levels in children with obesity and MetS. Using additional methods of analysis, such as multiple linear regression and mediation analyses, we aimed to investigate whether vitamin D may influence inflammation through these selected biomarkers, extending beyond simply reporting associations as performed in previous studies.

## 2. Results

### 2.1. Sample Characteristics and Epidemiology

A total of 80 children and adolescents were included, of which 36 (45.0%) were girls. The participants were aged 5 to 18 years, with a median age of 14 years and an interquartile range (IQR) of 4.0. Thirty children (13 girls) presented with body mass index (BMI) ≥5th to <85th percentile and were considered to have a normal weight, 5 children (4 girls) were in the overweight range (above the 85th percentile for age and sex) and 45 children (19 girls) were above 95th percentile and were considered to have obesity. All controls had normal BMI percentiles. The patient group was defined as all children and adolescents with excess weight (above the 85th percentile for age and sex). Out of 80 participants, 7 (8.8%) had MetS, of which 1 was a girl. All of them were in the patient group (BMI > 85th percentile).

Results are presented as median (IQR). Normality of the data was tested using the Kolmogorov–Smirnov and Shapiro–Wilk tests. We used the Mann–Whitney test as some of the variables were not normally distributed. Statistically significant differences between groups are marked with * (Table 1).

Regarding BMI (*p* < 0.001), waist circumference (WC) (*p* < 0.001), visceral fat thickness (VFT) (*p* < 0.001), and subcutaneous fat thickness (SFT) (*p* < 0.001), systolic (*p* < 0.001) and diastolic blood pressure (BP) (*p* < 0.001), children with obesity/overweight (B) exhibited higher values compared to children who are lean—controls (A). Mean values of triglycerides (TGC) (*p* = 0.003) were higher, and high-density lipoprotein cholesterol (HDL-C) levels were lower (*p* < 0.001) in patients (group B). Interestingly, fasting glucose values did not show significant differences between the groups (A; B; C). The WBC count was higher in the patient group with obesity/overweight (7.1 (3.5) × 10^9^/L) in comparison to the control group (5.7 (2.1) × 10^9^/L) (*p* = 0.013). Patients with MetS showed a trend toward a higher WBC count (*p* = 0.078) in comparison to controls. They had decreased adiponectin levels (*p* = 0.005), increased C-reactive protein (CRP) (*p* < 0.001), and myeloperoxidase (MPO) values (*p* = 0.019). Patients with obesity/overweight had a higher WBC count (*p* = 0.013), MPO (*p* = 0.006), and CRP (*p* = 0.002) values but only a trend towards decreased adiponectin levels (*p* = 0.086) and no significant difference between other measured inflammatory/oxidative stress markers in comparison to the control group. Superoxide dismutase-1 (SOD-1) is not reported in Tables since all values were under detection (62.5 pg/mL). Median values of vitamin D were significantly higher in the control group (A) (52.5 (19.2) nmol/L) in comparison to patient groups with obesity/overweight (B) (46.6 (16.3) nmol/L) (*p* < 0.001) and MetS (C) (38.1 (21.1) nmol/L) (*p* = 0.002).

### 2.2. Sample Characteristics According to Vitamin D Status

When comparing groups regarding vitamin D status (normo/hypovitaminosis D), statistical differences were found between BMI, WC, VFT, and adiponectin, which are parameters indicating abdominal obesity. Additionally, there was a significant difference between TGC levels in both groups (*p* = 0.042) (Table 2).

### 2.3. Correlations between Vitamin D Status, Visceral and Subcutaneous Fat Thickness, Metabolic Syndrome Parameters, and Inflammation/Oxidative Stress Parameters

The results of correlations between vitamin D and other parameters studied are presented in Figure 3. Additionally, Figure 4 shows a heat map of correlations for all variables. Due to the numerous correlations, a correction was calculated with the false discovery rate (FDR) method (Table 3). The correlations reaching the significance are marked.

After FDR correction, vitamin D was inversely correlated with BMI (*p* = 0.002), WC (*p* = 0.001), VFT (*p* = 0.009), and SFT (*p* = 0.019) and positively correlated with HDL-C values (*p* = 0.024). Higher systolic BP showed a trend toward (*p* = 0.075) lower vitamin D levels (Table 3).

Figure 4 presents a heat map depicting the correlations between vitamin D levels and various metabolic, inflammatory, and oxidative stress biomarkers in children with obesity and MetS. The heat map reveals a strong negative correlation between vitamin D levels and BMI (dark blue), as well as between WC, SFT, and VFT, suggesting that lower vitamin D levels are associated with higher BMI and abdominal obesity. Interestingly, no significant correlation was found between vitamin D and either traditional biomarkers or the novel biomarker I-TAC. These findings suggest that, while vitamin D is inversely associated with several obesity-related parameters, its influence on specific inflammatory pathways, such as those involving I-TAC, may be limited.

### 2.4. Multiple Linear Regression Analysis for Inflammation/Oxidative Stress Parameters as the Dependent Variable

Multiple linear regression analysis was performed with inflammation/oxidative stress parameters as a dependent variable and vitamin D and BMI as independent variables. The adjusted R^2^ was 0.0872 with *p* = 0.009 for obesity and *p* = 0.435 for vitamin D, which would mean that obesity is correlated with inflammation/oxidative stress and vitamin D levels are not. Coefficients were determined using a principal component analysis (PCA) method.

Inflammation/oxidative stress was defined by the following formula:Inflammation/oxidative stress = 0.739 × (CRP − 4.38)/4.41 + 0.759 × (WBC − 6.97)/2.41 + 0.667 × (MPO − 1651.96)/1597.32 − 0.351 × (ADIPONECTIN − 4988.63)/2177.66 + 0.529 × (I-TAC − 78.14)/60.088 + 0.1377 × (MCP-1 − 362.7)/92.165 − 0.0942 × (SOD-1 − 62.6)/62.5

### 2.5. Mediation Analysis for Inflammation/Oxidative Stress Parameters as the Dependent Variable

The mediation analysis evaluated whether vitamin D mediates the relationship between BMI and inflammation/oxidative stress. Our results show that vitamin D does not significantly mediate this relationship (Figure 5 and Figure 6).

*a*—this coefficient represents the effect of the independent variable (BMI) on the mediator variable (vitamin D). It indicates how changes in the BMI lead to changes in the vitamin D;*b*—this coefficient represents the effect of the mediator variable (vitamin D) on the dependent variable (inflammation and oxidative stress), controlling for the independent variable (BMI). It shows how changes in the mediator affect the outcome variable (inflammation and oxidative stress);*c*—this coefficient represents the total effect of the independent variable (BMI) on the dependent variable (inflammation and oxidative stress). It includes both the direct effect of BMI on inflammation and oxidative stress and the indirect effect that operates through the mediator vitamin D;*a × b*—mediation effect (illustrating how the independent variable influences the dependent variable through the mediator);*c*′—direct effect, which is the effect of the independent variable on the dependent variable when the mediator is included in the model.

Specifically, the indirect effect of BMI on inflammation/oxidative stress through vitamin D accounts for approximately 10% of the total effect, but this contribution is not statistically significant. Conversely, the direct effect of BMI on inflammation/oxidative stress remains substantial, at around 90%. This indicates that while BMI has a strong direct impact on inflammation/oxidative stress, vitamin D does not substantially influence this relationship. Similar results were observed in children with obesity and MetS as an independent variable, suggesting that addressing vitamin D levels alone may not significantly alter inflammation or oxidative stress associated with high BMI.

## 3. Discussion

This cross-sectional study explored the links between vitamin D, inflammation, oxidative stress, and some features of MetS, including abdominal obesity, in the pediatric population. We observed increased markers of inflammation and oxidative stress and decreased levels of vitamin D in pediatric obesity but no direct correlation between selected biomarkers of inflammation and oxidative stress and vitamin D status.

### 3.1. Metabolic Syndrome, Obesity, Inflammation and Oxidative Stress

An important characteristic of MetS, besides altered glucose metabolism with increased risk of developing type 2 diabetes [27], arterial hypertension, and dyslipidemia, is abdominal obesity [4,28]. Excess weight, especially in the abdominal part, is associated with increased secretion of adipokines and inflammatory cytokines [29]. Specifically, obesity is thought to be related to changes in adipokine secretion, the activity of adipose tissue macrophages, helper T cells, and regulatory T cells [30], as well as to adipogenesis, lipid metabolism, and thermogenesis [31].

Another characteristic of MetS is arterial hypertension [4,28]. Children with increased cardiovascular risk have an impaired antioxidants/oxidative stress balance such as reduced endogenous levels of vitamins C and E, reduced glutathione, increased levels of malondialdehyde and oxidized low-density lipoproteins as well as decreased levels of superoxide dismutase and decreased total antioxidant capacity [32]. Interestingly, the total WBC count and neutrophil parameters alone were found to have a positive predictive value in estimating the degree of cardiovascular risk among individuals with obesity [33].

Our results indicate a trend toward an increased WBC count in patients with MetS. They also exhibited higher levels of CRP and MPO, and lower levels of adiponectin. Interestingly, I-TAC levels were not altered in children with obesity or MetS. Elevated I-TAC levels have been observed in adults with long-term obesity and associated comorbidities [13,14], suggesting that I-TAC elevation may primarily occur in more severe or chronic stages of obesity. In children, the duration and severity of obesity may not yet have reached a level sufficient to significantly elevate I-TAC. The adult studies included patients with varying levels of metabolic health and comorbidities, which could have contributed to the higher I-TAC levels not observed in our pediatric cohort.

Children with obesity/overweight had higher systolic and diastolic BP, as well as parameters of abdominal obesity, which are WC, VFT, and SFT. They also had a higher WBC count and CRP and MPO levels, indicating increased abdominal adiposity and inflammation/oxidative stress in children and adolescents with overweight/obesity. This is consistent with a study in young adults that showed that total leukocyte and monocyte counts were significantly increased in obesity. Total leukocyte count was associated with liver enzyme levels, insulin resistance, VFT and SFT, neutrophil count with insulin resistance, lymphocyte count with serum liver enzymes, insulin resistance, and dyslipidemia, and monocyte count with serum liver enzymes, insulin resistance, VFT, SFT, body fat mass, and body fat percentage. The study concluded that chronic low-grade systemic inflammation in young adults with obesity is evident and is associated with obesity-related complications such as non-alcoholic fatty liver disease, insulin resistance, and dyslipidemia [34]. Also, in children and adolescents, the findings show that already a routine laboratory parameter, such as total leukocyte count, may be a reliable inflammation-related marker in patients with obesity, even without MetS [35].

### 3.2. Obesity, Vitamin D Status, Inflammation and Oxidative Stress

Pre-obesity and obesity in childhood have been shown to be positively associated with inflammation as well as oxidative stress [28,36]. There is also an inverse association between pre-obesity, obesity, and vitamin D status [36].

Several studies have found an inverse relationship between human plasma 25-hydroxyvitamin (25(OH)) D concentration and obesity [37]. Different mechanisms have been tested to explain the correlation between hypovitaminosis D and obesity, such as lesser skin exposure to sunlight, lower dietary intake, decreased intestinal absorption, impaired hydroxylation in adipose tissue and 25(OH)D accumulation in fat, and low-grade inflammation [24,38]. The volume dilution theory postulates that serum vitamin D levels decrease with increasing body size. The theory of sequestration of vitamin D in adipose tissue, ingested or synthesized vitamin D speculates that it becomes tightly bound in fat stores and does not enter the bloodstream in sufficient quantities to maintain serum levels. Other theories suggest that individuals who are lean have a better ability to activate vitamin D in adipose tissue than those who are obese and that in obesity, there is a lower expression of the enzymes 1α-hydroxylase (mitochondrial CYP27B1) and 25-hydroxylase (CYP2J2) in adipocytes compared to people who are lean [24,39,40,41,42].

A plausible explanation is that a volume dilution or sequestration of vitamin D in excess fat is a major reason for hypovitaminosis D in obesity [24,39,40,41,42]. This explanation is concordant with our results, which showed that vitamin D was inversely correlated to BMI, WC, VFT, and SFT, indicating the importance of excess fat, especially abdominal fat, in lower vitamin D levels. This is in line with several other studies that report an inverse relationship between vitamin D levels and obesity [36,37]. Furthermore, in reported studies, increased BMI was correlated with increased CRP and total antioxidant status, whereas vitamin D level was decreased. However, correlation was reported between obesity and inflammation/oxidative stress but not directly to vitamin D status [36].

A recent systematic review reported mixed results regarding inflammation, oxidative stress, and vitamin D status in children and adolescents. They found an association between vitamin D status and biomarkers of oxidative stress and inflammation such as CRP, interleukin-6, cathepsin S, vascular cell adhesion molecule-1, malondialdehyde, myeloperoxidase, 3-nitrotyrosine, and superoxide dismutase, but only in five out of eight studies. Some of them adjusted models for confounding factors in obesity (such as BMI and body fat thickness) [43].

According to our results, hypovitaminosis D (independently of BMI status) was associated only with increased adiponectin levels and not with other biomarkers of inflammation and oxidative stress (WBC counts, CRP, MCP-1, MPO, I-TAC, SOD-1). However, when adjusting for results of multiple correlations between vitamin D, inflammation/oxidative stress markers and other measured parameters, also adiponectin failed to be significantly correlated with vitamin D levels. Also, multiple linear regression showed no connection between vitamin D levels and inflammatory/oxidative stress parameters but only between obesity and increased inflammation/oxidative stress. In addition, mediation analyses did not reveal a significant mediating role for vitamin D in the relationship between BMI and inflammation/oxidative stress. Instead, the elevated biomarkers of oxidative stress and inflammation observed in obesity may be more closely related to obesity itself rather than to low vitamin D levels.

### 3.3. Obesity, Vitamin D Supplementation, Inflammation and Oxidative Stress

In our study, no correlation between markers of inflammation, oxidative stress, and vitamin D status was seen. This is in opposition to the observed beneficial effects of vitamin D supplementation in the pediatric population with overweight and obesity, where authors concluded that vitamin D exerts an anti-inflammatory effect via decreasing the CRP levels and protecting stable values of IL-10 with less effect on pro-inflammatory factors such as lL-17 and leukocyte profile parameters [30]. However, our findings are concordant with results from a recent observational and Mendelian randomization study, where authors could not prove the beneficial role of vitamin D supplementation on obesity-related inflammation [23]. According to our results, the effectiveness of standard vitamin D supplementation in reducing inflammation and oxidative stress in children with obesity is questionable. However, our results should be validated in a prospective study including vitamin D supplementation.

### 3.4. Limitations

Seasonal variations in vitamin D levels due to sun exposure were addressed by avoiding measurements during summer, though some residual variability may remain. Despite excluding participants using vitamin D supplements and considering dietary specifics through anamnesis, residual variability in vitamin D levels due to diet may remain. Additionally, while our groups were balanced according to gender distribution, potential residual effects related to sex differences in inflammation/oxidative stress and vitamin D levels cannot be entirely ruled out. The study’s participant population, predominantly Caucasian, might limit the generalizability of our findings to other ethnic groups, as ethnic differences in vitamin D metabolism and inflammatory responses could influence the observed relationships. Issues with biomarker measurement accuracy and assay reliability might have impacted our results, as the specific methods used could introduce variability. Additionally, the interaction between vitamin D and inflammatory processes may involve complex biological mechanisms not fully captured by the biomarkers selected for our study.

The small sample size may affect the generalizability and statistical power of our findings despite our efforts in sample size estimation calculations. Given the cross-sectional design, which cannot establish causality, we conducted additional analyses, including mediation and multiple linear regression analyses. The effectiveness of mediation analysis depends on the robustness of the statistical methods employed. Despite controlling for various confounders, unmeasured variables could still affect the observed relationships. One limitation of this study is the relatively small number of children who are severely obese, which may reduce the statistical power to detect significant associations. Furthermore, we did not consider the duration of obesity, which could affect our results by failing to capture the cumulative effects of prolonged obesity on health outcomes.

Future research should consider larger and more diverse cohorts to enhance statistical power and generalizability. Longitudinal studies could provide insights into causal relationships by examining how changes in vitamin D levels over time influence inflammation and oxidative stress. Incorporating a broader range of biomarkers and exploring other potential mediators, such as genetic factors or lifestyle variables, may offer a more comprehensive understanding of the interactions between vitamin D, obesity, and inflammation/oxidative stress. Investigating the effects of vitamin D supplementation in randomized controlled trials could further elucidate its role and potential therapeutic benefits in managing obesity and related inflammatory conditions.

## 4. Materials and Methods

### 4.1. Study Description

We performed a prospective cross-sectional study to investigate associations between vitamin D levels, inflammation, oxidative stress, abdominal obesity, and other features of MetS in children and adolescents. The ethical aspects of the Helsinki Declaration were followed (Association, 2013). The study methods and procedures were approved by the institutional ethics committee (The Medical Ethics Committee of the University Medical Centre Maribor) and the national ethics committee (The National Medical Ethics Committee of the Republic of Slovenia).

### 4.2. Sample

The sample size estimation was performed using an online tool from the Cleveland Clinic site, according to a published article [44], utilizing an alpha value of 0.05 and a test power of 80%. Eighty-two children and adolescents aged 5 to 18 were recruited from ambulatory visits in University Medical Centre Maribor, Slovenia, Department of Pediatrics. All participants were screened for confounding factors, including regular therapy, vitamin D supplementation, and comorbid states, which could influence cardiometabolic health indices. These factors were included as exclusion criteria. Two participants were excluded due necessity of regular therapy intake, which was not reported at the time of patient screening. Written informed consent was obtained from the parents or legal guardians of each participant or the participants themselves if aged above 15. Participants were assured of the confidentiality and voluntary nature of their participation.

### 4.3. Data Collection

Data were gathered from the end of October 2022 to the beginning of May 2023. Fasting venous blood samples were collected for assessment of 25-OH vitamin D levels, triglycerides, high-density cholesterol, fasting glucose levels, C-reactive protein, white blood cell count (leukocytes) and selected biomarkers of chronic systemic inflammation/oxidative stress which were adiponectin, monocyte chemoattractant protein-1, myeloperoxidase, interferon-inducible T-cell alpha chemoattractant and superoxide dismutase-1. They were measured with the use of enzyme-linked immunosorbent assay (ELISA) kits. Systemic blood pressure was measured using a digital automatic device (Omron^®^; Osaka, Japan) on a single occasion in a sitting position after 5 min of rest. Abdominal ultrasound was performed by trained physicians using an abdominal transducer to measure visceral and subcutaneous fat thickness. Measurements were performed according to the Ultrasound protocol for visceral fat and abdominal subcutaneous fat in both adults and young children [45,46]. The visceral fat thickness was defined as the distance between the peritoneum and the corpus of the lumbar vertebra, and the subcutaneous fat measurement as the distance between the skin and the ventral edge of the abdominal muscles (linea alba).

Body mass (kg), body height (cm), and waist circumferences (cm) measurements were assessed following World Health Organization guidelines [47]. Waist circumference was assessed with a tape with an accuracy of 0.1 cm. A stadiometer attached to the scale was used to measure body height. Body mass was measured using a digital scale with 0.1 kg accuracy. Body mass index and the corresponding percentiles were calculated according to the Centers for Disease Control and Prevention Clinical growth charts [48].

Children who presented with BMI ≥ 5th and <85th percentile for age and sex were considered to have a normal weight, those with BMI ≥ 85th percentile for age and sex were considered to be in an overweight range, and those with BMI ≥ 95th percentile for age and sex were considered to have obesity [48,49].

MetS in children aged 10 to 16 years was defined according to The International Diabetes Federation (IDF) definition [50]:Waist circumference ≥90th percentile* AND;Number of abnormalities ≥2;Triglyceride ≥150 mg/dL (1.7 mmol/L);High-density lipoprotein cholesterol <40 mg/dL (1.03 mmol/L);Blood pressure either:Systolic >130 mmHg;Diastolic ≥85 mmHg;Glucose ≥100 mg/dL (5.6 mmol/L).

* Ethnic-specific waist circumference [51]

For children 16 years and older, the adult criteria were used. In comparison to the adults’ definition of MetS, the definition for children aged 10 to 16 years uses ethnic-specific waist circumference percentiles and one cut-off level for high-density lipoprotein rather than a sex-specific cut-off. For children younger than 10 years of age, MetS cannot be diagnosed. However, caution is still recommended if the waist circumference is ≥90th percentile [52,53,54].

Severe vitamin D deficiency is defined as a serum 25(OH)D concentration below <30 nmol/L (or 12 ng/mL) and dramatically increases the risk of excess mortality, infections, and many other diseases, and should be avoided whenever possible. Vitamin D deficiency is defined as serum 25-hydroxyvitamin D [25(OH)D] < 50 nmol/L (or 20 ng/mL). These levels are associated with unfavorable skeletal outcomes, including fractures and bone loss. A 25(OH)D level of >50 nmol/L or 20 ng/mL is considered sufficient, although some data suggest a benefit for a higher threshold [55].

For analysis, we divided participants into two main categories according to vitamin D levels: the vitamin D insufficient group (serum levels less than 50 nmol/L) and the vitamin D sufficient group (serum levels above 50 nmol/L) according to the recommendation [55].

### 4.4. Data Management and Statistical Analysis

Statistical analyses were performed using IBM SPSS version 29, R Statistics, and Jamovi. Descriptive statistics were used to summarize demographic characteristics. Spearman’s rho correlation analysis was conducted to explore associations between 25(OH) vitamin D, triglycerides, high-density cholesterol, fasting glucose, C-reactive protein, white blood cell count, adiponectin, monocyte chemoattractant protein-1, myeloperoxidase, interferon-inducible T-cell alpha chemoattractant, superoxide dismutase-1, body mass index, waist circumference, visceral and subcutaneous fat thickness. Correction for multiple correlations in Spearman’s rho correlation analysis was calculated using the False Discovery Rate method. Multiple linear regression analysis was performed for inflammation/oxidative stress parameters as the dependent variable and vitamin D levels and BMI (BMI ≥ 5th and <85th percentile for controls, ≥ 85th percentile for age and sex for patients with obesity/overweight) as the independent variables. Mediation analysis was used to evaluate whether vitamin D mediates the relationship between BMI and inflammation/oxidative stress.

## 5. Conclusions

Our study investigated the relationship between vitamin D levels, obesity, and markers of inflammation and oxidative stress in children and adolescents with obesity and/or MetS. Despite utilizing thorough analyses, including mediation and multiple linear regression, we did not find a significant mediating effect of vitamin D on the relationship between BMI and inflammation/oxidative stress. This suggests that while lower vitamin D levels are associated with higher BMI and increased abdominal fat, the observed inflammatory and oxidative stress biomarkers may be more closely linked to obesity rather than directly influenced by low vitamin D levels. Moreover, the novel biomarker I-TAC, which has been examined in adult obesity research, did not show significant changes in our pediatric population. This finding implies that its clinical relevance in childhood obesity may be limited. Further research is needed to better understand these relationships and explore additional factors that might contribute to the complex interactions between vitamin D, obesity, MetS, and inflammation/oxidative stress.

## Figures and Tables

**Figure 1 ijms-25-10599-f001:**
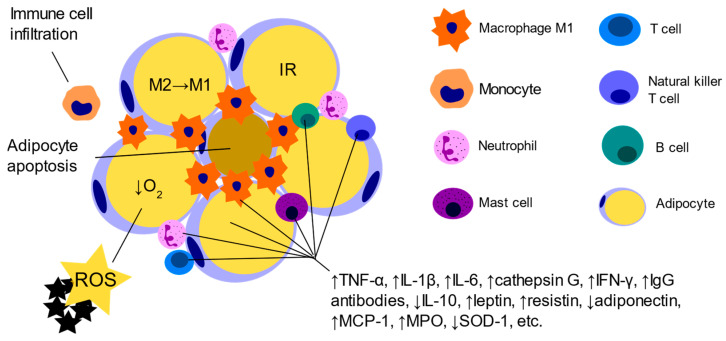
Inflammation and oxidative stress in obese adipose tissue. Dysfunctional adipocytes secrete pro-inflammatory adipokines and recruit immune cells. Activated T cells, along with adipocyte-derived chemoattractants, facilitate the migration of monocytes into adipose tissue, where they differentiate from anti-inflammatory macrophages (M2) into pro-inflammatory macrophages (M1). Hypertrophic adipocytes and resident immune cells downregulate the secretion of anti-inflammatory cytokines while upregulating the release of inflammatory adipokines and cytokines, contributing to peripheral insulin resistance (IR). The hypertrophy of adipocytes leads to hypoxia (↓O2), which is associated with the production of reactive oxygen species (ROS), resulting in increased oxidative stress and the maintenance of chronic low-grade inflammation [11,12]. TNF-α—tumor necrosis factor-alpha, IL-1β—interleukin-1 beta, IL-6—interleukin 6, IFN-γ—interferon gamma, IgG—immunoglobulin G, IL-10—interleukin 10, MCP-1—monocyte chemoattractant protein-1, MPO—myeloperoxidase, SOD-1—superoxide dismutase-1.

**Figure 2 ijms-25-10599-f002:**
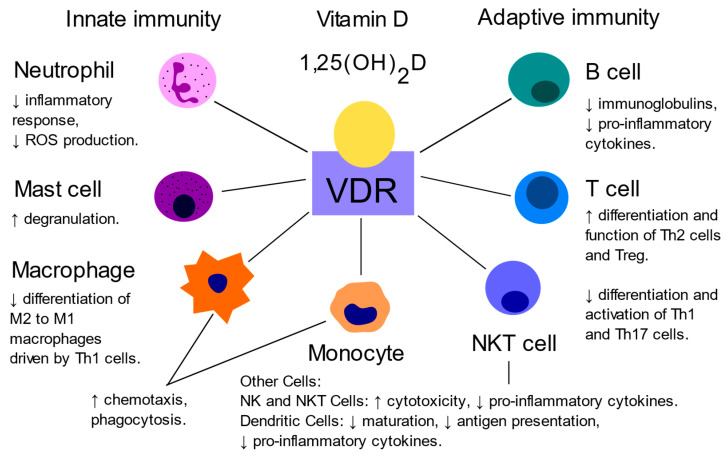
Immune cells present in obese adipose tissue expressing vitamin D receptor (VDR). Vitamin D exerts its immunoregulatory effects by binding to VDR, which is expressed in various immune cells. In the innate immune system, vitamin D enhances the chemotaxis and phagocytic activity of monocytes and macrophages, promotes the degranulation of mast cells, and modulates the activity of dendritic and natural killer (NK) cells. In the adaptive immune system, vitamin D facilitates the differentiation and function of T-helper 2 (Th2) cells and regulatory T cells (Treg) while concurrently suppressing the differentiation and activation of T-helper 1 (Th1) and T-helper 17 (Th17) cells. Additionally, vitamin D also modulates B cell function and immunoglobulin production [19,25].

**Figure 3 ijms-25-10599-f003:**
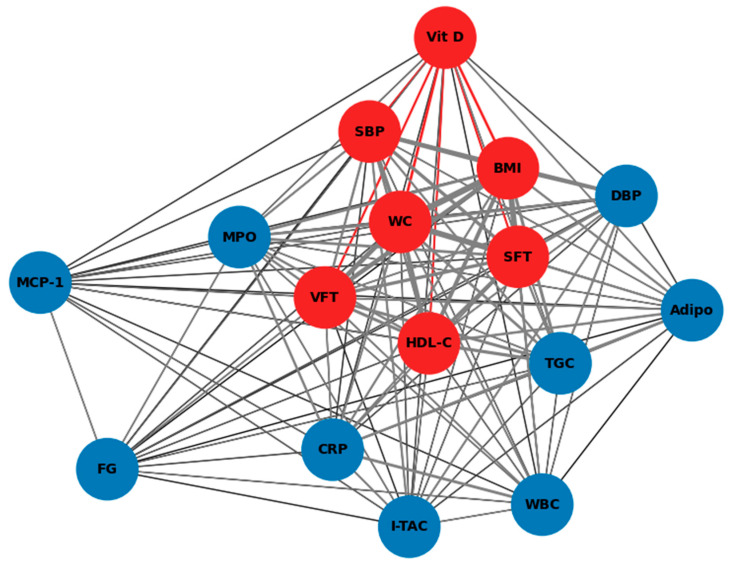
Spearman correlations network between vitamin D status, abdominal fat thickness, metabolic syndrome parameters, and inflammation/oxidative stress parameters. Statistically significant correlations with vitamin D are represented by red dots and red lines. Blue dots represent other parameters that are not significantly correlated with vitamin D. Vit D—vitamin D, BMI—body mass index, WC—waist circumference, SFT—subcutaneous fat thickness, VFT—visceral fat thickness, HDL-C—high-density lipoprotein cholesterol, SBP—systolic blood pressure, DBP—diastolic blood pressure, TGC—triglycerides, FG—fasting glucose, Adipo—adiponectin, CRP—C-reactive protein, MPO—myeloperoxidase, MCP-1—monocyte chemoattractant protein-1, WBC—white blood cells, I-TAC—interferon-inducible T-cell alpha chemoattractant.

**Figure 4 ijms-25-10599-f004:**
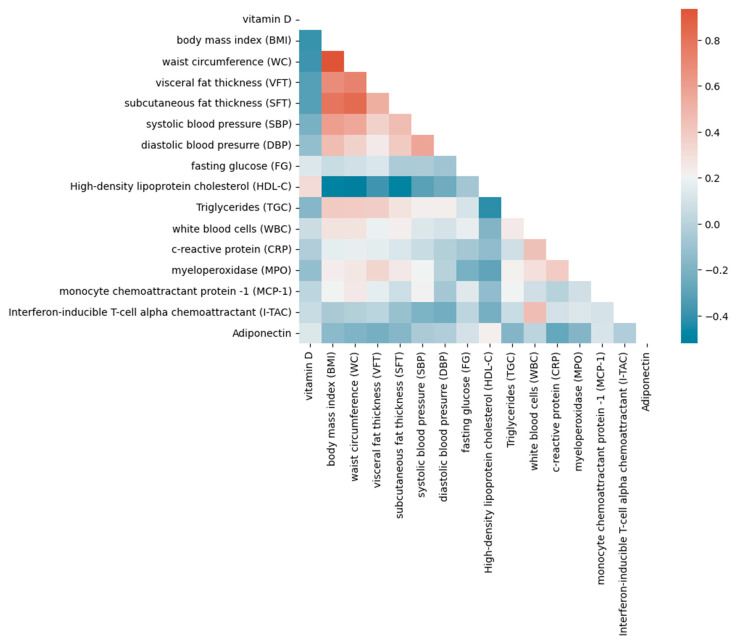
Heat map of Spearman correlations between vitamin D status, abdominal fat thickness, metabolic syndrome parameters, and inflammation/oxidative stress parameters. Positive correlations are represented by shades of red, while negative correlations are indicated by shades of blue.

**Figure 5 ijms-25-10599-f005:**
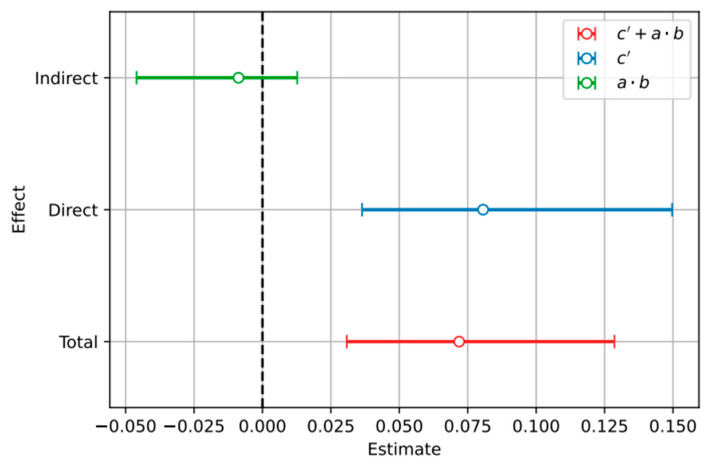
Estimation plot of mediation analysis between BMI as the independent variable, vitamin D as the mediator variable, and inflammation and oxidative stress as the dependent variable. On the mediation plot, the coefficients *a*, *b*, and *c*′ are displayed along with their standard errors. It is evident that the indirect path (coefficient *a* × *b*) is statistically non-significant, as it intersects the value of 0 (dashed line).

**Figure 6 ijms-25-10599-f006:**
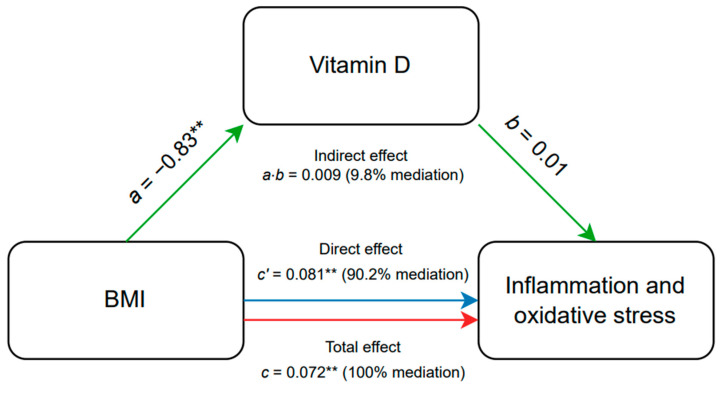
Mediation path plot of analysis between BMI as the independent variable, vitamin D as the mediator variable, and inflammation and oxidative stress as the dependent variable. ** *p* < 0.01. BMI—Body Mass Index.

**Table 1 ijms-25-10599-t001:** Descriptive statistics of metabolic syndrome (MetS) parameters, abdominal fat thickness, and inflammation/oxidative stress parameters with results of the Mann–Whitney test between controls, patients with obesity/overweight, and patients with MetS.

Parameters	Controls, Median (IQR), N = 30, A	Patients with Obesity, Overweight, Median (IQR), N = 80, B	Patients with MetS, Median (IQR), N = 7, C	*p*-Values (A and B)	*p*-Values (A and C)
Body mass index (kg/m^2^)	19.9 (2.9)	30.2 (6.9)	32.7 (4.6)	<0.001 *	< 0.001 *
Waist circumference (cm)	70.5 (5.8)	95.0 (20.5)	112.0 (7.0)	<0.001 *	< 0.001 *
Visceral fat thickness (mm)	36.8 (11.9)	60.0 (23.8)	64.0 (35.7)	<0.001 *	< 0.001 *
Subcutaneous fat thickness (mm)	10.1 (10.7)	36.5 (17.0)	45.0 (26.5)	<0.001 *	< 0.001 *
Systolic blood pressure (mmHg)	114.0 (19.5)	128.0 (21.3)	129.0 (19.0)	<0.001 *	0.009 *
Diastolic blood pressure (mmHg)	68.0 (10.5)	80.5 (13.0)	78.0 (14.0)	<0.001 *	0.062
Fasting glucose (mmol/L)	4.6 (0.5)	4.6 (0.4)	4.5 (0.7)	= 0.928	0.785
High-density lipoprotein cholesterol (mmol/L)	1.6 (0.5)	1.2 (0.3)	0.9 (0.3)	<0.001 *	< 0.001 *
Triglycerides (mmol/L)	0.7 (0.5)	1.1 (0.8)	1.9 (0.8)	0.003 *	< 0.001 *
White blood cells (×109/L)	5.7 (2.1)	7.1 (3.5)	7.7 (3.7)	0.013 *	0.078
C-reactive protein (mg/L)	3 (0)	3 (0)	3 (5.5)	0.002 *	< 0.001*
Myeloperoxidase (ng/mL)	290.0 (2778.8)	1432.5 (2877.0)	3615.0 (1465.0)	0.006 *	0.019 *
Monocyte chemoattractant protein-1 (pg/mL)	360.8 (103.0)	333.0 (97.0)	456.0 (270.0)	0.633	0.358
Interferon-inducible T-cell alpha chemoattractant (pg/mL)	62.5 (0.0)	62.5 (0.0)	62.5 (0.0)	0.738	0.922
Adiponectin (ng/mL)	5354.7 (1730.0)	4190.0 (2962.5)	3250.0 (2740.0)	0.086	0.005 *
Vitamin D (nmol/L)	52.8 (19.2)	46.6 (16.3)	38.1 (21.1)	<0.001 *	0.002 *

* Statistically significant difference.

**Table 2 ijms-25-10599-t002:** Descriptive statistics and Mann–Whitney test for vitamin D status and metabolic syndrome parameters, abdominal fat thickness, and inflammation/oxidative stress parameters.

Parameters	Normovitaminosis D, Median (IQR), N = 41	Hypovitaminosis D, Median (IQR), N = 39	*p*-Values
Body mass index (kg/m^2^)	23.3 (10.2)	25.6 (13.0)	0.024 *
Waist circumference (cm)	83.0 (25.0)	92.0 (31.0)	0.022 *
Visceral fat thickness (mm)	46.7 (26.8)	55.5 (26.8)	0.035 *
Subcutaneous fat thickness (mm)	20.6 (29.3)	31.5 (25.5)	0.115
Systolic blood pressure (mmHg)	117.5 (21.0)	127.0 (23.0)	0.063
Diastolic blood pressure (mmHg)	74.0 (17.0)	78.0 (17.0)	0.221
Fasting glucose (mmol/L)	4.8 (0.5)	4.5 (0.4)	0.073
High-density lipoprotein cholesterol (mmol/L)	1.4 (0.6)	1.2 (0.5)	0.115
Triglycerides (mmol/L)	0.7 (0.8)	1.0 (0.7)	0.042 *
White blood cells (×109/L)	6.2 (3.6)	6.8 (3.0)	0.969
C-reactive protein (mg/L)	3.0 (0.0)	3.0 (0,0)	0.833
Myeloperoxidase (ng/mL)	785.0 (2850.0)	1542.5 (2966.3)	0.494
Monocyte chemoattractant protein-1 (pg/mL)	348.0 (88.0)	333.0 (157.5)	0.364
Interferon-inducible T-cell alpha chemoattractant (pg/mL)	62.5 (0.0)	62.5 (0.0)	0.105
Adiponectin (ng/mL)	5195.0 (2672.5)	4190.0 (2832.5)	0.032 *
Vitamin D (nmol/L)	58.4 (16.1)	39.5 (10.9)	<0.001 *

* Statistically significant difference.

**Table 3 ijms-25-10599-t003:** Correlations between vitamin D status, abdominal fat thickness, metabolic syndrome parameters, and inflammation/oxidative stress parameters with the use of Spearman’s correlation coefficient.

	Vitamin D (nmol/L)
Body mass index (kg/m^2^)	r = −0.385 *p* = 0.002 *
Waist circumference (cm)	r = −0.402 *p* = 0.001 *
Visceral fat thickness (mm)	r = −0.329 *p* = 0.009 *
Subcutaneous fat thickness (mm)	r = −0.300 *p* = 0.019 *
Systolic blood pressure (mmHg)	r = −0.234 *p* = 0.075
Diastolic blood pressure (mmHg)	r = −0.100 *p* = 0.489
Fasting glucose (mmol/L)	r = 0.111 *p* = 0.444
High-density lipoprotein cholesterol (mmol/L)	r = 0.289 *p* = 0.024 *
Triglycerides (mmol/L)	r = −0.213 *p* = 0.111
White blood cells (×109/L)	r = 0.034 *p* = 0.804
C-reactive protein (mg/L)	r = −0.099 *p* = 0.499
Myeloperoxidase (ng/mL)	r = −0.153 *p* = 0.282
Monocyte chemoattractant protein-1 (pg/mL)	r = 0.053 *p* = 0.710
Interferon-inducible T-cell alpha chemoattractant (pg/mL)	r = 0.091 *p* = 0.517
Adiponectin (ng/mL)	r = 0.024 *p* = 0.124

* Statistically significant difference after False Discovery Rate Method correction.

## Data Availability

The raw data supporting the conclusions of this article will be made available by the authors on request.

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
