# Peer review of "Metabolic Syndrome, Inflammation, Oxidative Stress, and Vitamin D Levels in Children and Adolescents with Obesity"

_ijms, 2024, doi:10.3390/ijms251910599_

Round 1

Reviewer 1 Report

Comments and Suggestions for Authors

Petek et al. reported the correlation between vitamin D levels and obesity, metabolic syndrome, oxidative stress, and other metabolic parameters. The findings are interesting and may provide a valuable clinical insight regarding the importance of vitamin D for metabolic health.

I would like to address following points to improve the manuscript:

1. In the method section, authors wrote that the samples were collected from subjects who visited University Medical Centre, Department of Pediatrics. Please clarify the city and country of this sample collection site.

2. The dietary intake of the subjects may influence the vitamin D level, especially if they take vitamin D supplements. Please provide information for this point.

3. For Table 3, providing graph to show correlation status between vitamin D and some parameters, especially for those that showed significance, is advised. The visualization of the data can be elevated rather than showing in the table.

4. Since the authors did not measure the levels of TNF and interleukins in this study, the inclusion of Figure 1 at the beginning of the discussion may not be entirely relevant to the results presented. It would be advisable to reconsider the placement or context of Figure 1 to ensure it aligns more closely with the study's findings.

Comments on the Quality of English Language

Moderate editing is required.

Reviewer 2 Report

Comments and Suggestions for Authors

Metabolic syndrome is associated with systemic inflammation, oxidative stress, and hypovitaminosis D. Although an association between inflammation and obesity is evident, a relationship between vitamin D levels and inflammation in this process is not clear. This study aimed to investigate this relationship in children with obesity and MetS. They found no link between vitamin D and inflammatory/oxidative stress variables. While the study is solid based on data from 80 patients and correlation and multivariate analyses, it is not immediately clear what are the unique findings beyond the literature. It is recommended that some important points be addressed and clarified.

  1. Two major conclusions have been drawn in this study. One is that there is no direct correlation of the markers of inflammation and oxidative stress with the vitamin D status. The other is the finding of increased inflammation/oxidative stress and decreased vitamin D levels in both obesity and metabolic syndrome. 

As introduced by the authors, these two findings have been reported by others through various studies. I would recommend that the authors make it clear what is unique in this study, whether the choice of cohort, stratification of the samples, or methods of analysis. 

As pointed out by the authors, the relationship between vitamin D levels and inflammation in obese patients has been controversial. It would be necessary to discuss what factors could have led to this discrepancy of reports and what makes the current study unique, and what needs to be further addressed in this regard in the future.

  1. The authors found that “Our results indicate that vitamin D levels are lower in children with obesity and/or metabolic syndrome”. This seems to have been widely observed before, based on the literature cited. It is not clear whether this study simply repeats the existing findings or something novel can be concluded by the authors.
  2. Is gender considered as a factor? It is not evident in the manuscript.
  3. The manuscript reads like having multiple aims and the conclusions are mingled. Again, the manuscript would be greatly improved if the novelty of this study could stand out on the basis of solid and clear data presentation.
